# The Value of Targeting Complement Components in Asthma

**DOI:** 10.3390/medicina56080405

**Published:** 2020-08-12

**Authors:** Marwa M. E. Mohamed, Alicia D. Nicklin, Cordula M. Stover

**Affiliations:** Department of Respiratory Sciences, University of Leicester, Leicester LE1 9HN, UK; mmem3@leicester.ac.uk (M.M.E.M.); an300@student.le.ac.uk (A.D.N.)

**Keywords:** asthma, complement, treatment

## Abstract

Asthma is an important respiratory illness. Though pharmacological and biological treatment is well established and is staged according to endotypes and their responses to treatment, novel avenues are being explored. Our focus is complement. In this viewpoint, we evaluate the approach to target complement in this complex hypersensitivity reaction that develops chronicity and has a personal—as well as a societal—cost.

## 1. Background

Since 2008 till present, the Word Health Organization estimates that the global prevalence of asthma has increased by another 39 million people—totaling 339 million. It is a potentially fatal restrictive respiratory disease with most deaths occurring in the developing world [1]. In the developed world, asthma remains a noncommunicable disease of epidemiological importance [2]. It is therefore essential to deepen our understanding of the pathophysiology of asthma with an aim to relieve the disease burden, which affects a substantial proportion of the world’s population.

Genetic susceptibilities, exposure to airborne particulate matter and inhalation of potential immunogens shape the development of an allergic response that lead to a state of airway hyperresponsiveness. Complement activation products C3a and C5a, which exert their function by binding to their receptors C3aR and C5aR, are thought to modulate the response to allergens via influencing activities of cells involved in the innate and adaptive response [3]. In susceptible persons, bronchoscopy assisted segmental challenge with specific allergens produced increased C3a and C5a in the bronchoalveolar lavages [4]. C3a levels were significantly elevated in patients with exacerbations of their allergic asthma and seemed to discriminate patients’ responsiveness to standard treatment [5]. Certain allergens—such as those associated with house dust mites—display natural protease activity which has been implicated to cleave complement components directly [6]. Early activation of complement may explain how environmental triggers can directly influence cell reactivities and alter susceptibilities, in short, favor the development of atopy [7].

The aim of this viewpoint article is to explore the rationale of targeting complement in the treatment of asthma.

## 2. Pathogenesis Driven Treatment

Allergic asthma is a type I hypersensitivity reaction, an exaggerated immune response of the lower respiratory tract to otherwise innocuous antigens, deemed allergens. Upon first exposure to allergens, antigen presenting cells are able to uptake these and interact with naïve T cells, subsequently inducing their activation [8]. Signals promote T-helper 2 (Th2) over T-helper 1 (Th1) polarization, a key cell involved in allergic asthma that provides major sources of the type 2 cytokines, IL-4, IL-5, IL-9 and IL-13 [9]. Major importance of IL-4 and IL-13 is immunoglobulin class switching in B cells, resulting in IgE production. IgE binds to high affinity receptors (FcεRI) on mast cells and basophils during the asymptomatic sensitization phase. Mast cell and basophil degranulation occur upon subsequent allergen exposures when allergens bind to membrane bound IgE, resulting in FcεRI cross linking and thus the extracellular release of proinflammatory mediators from preformed stores.

Preformed mediators released include histamine, heparin and neutral proteases (chymase and tryptase) while de novo synthesized mediators are predominantly eicosanoids, cytokines and chemokines [10]. Immediate asthmatic responses, which occur within minutes, involve vasodilation and bronchoconstriction, and subsequently the symptoms of shortness of breath, pain and wheeze. Cytokines, chemokines and eicosanoids exert numerous immediate and delayed responses. Most notably, they increase migration of immune cells to the lungs along a chemogradient. Attracted cells, including eosinophils, neutrophils and lymphocytes, in turn release inflammatory mediators that direct the pathogenesis of allergic asthma [11]. Of interest, IL-4, IL-5 and IL-6 are involved in B cell proliferation, immunoglobulin class switching (hence acting in a positive loop mechanism), eosinophil recruitment and survival [12]. In addition, IL-13, chymase and tryptase contribute to airway remodeling that increase mucus secretion, epithelial damage and collagen deposition, exacerbating asthma symptoms [13].

Complement has a role to play in both the immediate and time delayed reactions of type I hypersensitivity. Via the production of the anaphylatoxins C3a and C5a, complement is involved in the recruitment of inflammatory cells, activation of effector cells including airway smooth muscle contraction and hypersecretion of mucus from goblet cells [14]. The immediate phase of an asthmatic attack is mediated by readily available or prestored inflammatory mediators [15] and complement, which is activated as an innate immune defense system [14]. The delayed bronchoconstriction after the initial attack is driven by upregulated cytokines and ensuing cell–cell interactions. The role of complement 24 h after experimental challenge was, however, large enough to significantly contribute to the levels of IL-4, a Th2 cytokine, in bronchoalveolar lavages of a mouse model of *Aspergillus fumigatus* induced respiratory hypersensitivity [16]. Based on preclinical models, C3a and C5a are thought to have distinct roles to play in the sensitization and challenge phases of allergy [17], which may reflect their roles in the macrophage polarization in these phases [18].

Chronicity leads to structural change. Complement activation sustains an environment in which secretion of profibrotic growth factors and cytokines leads to airway smooth muscle hypertrophy, mucus gland hyperplasia and subepithelial collagen deposition [19]. C3aR and C5aR are described for mast cells, eosinophils, basophils, neutrophils, macrophages (cells of the myeloid lineage), for effector T cells and activated B cells (cells of the lymphoid lineage) and for bronchial epithelial cells, airway smooth muscle cells [20] as well as myofibroblasts [21]. A direct role of C3aR expressing mast cells on allergy induced proliferation of airway smooth muscle cells is likely [22]. The relative importance of complement for the acute and chronic phases of inflammation presumably differs. It is the chronic stage that is difficult to model in experimental animals yet is pertinent for the stage in which novel therapeutics are considered.

It is a clinical aim to target asthmatic inflammation and thereby reduce development of airway fibrosis [19], a feature of airway remodeling in chronic asthma [23]. Anti-eosinophil biologics have been developed to reduce the eosinophilic component of asthma when poorly controlled by inhaled corticosteroids (anti-IL-5 or IL5R monoclonal antibodies, mAbs) and to target Th2 cytokines (anti-IL-4/IL-13 mAbs) [24].

Approaches in targeting complement and its activation as well as a strategic appraisal have been previously presented [25]. C1 Inhibitor infusion prior to intrabronchial house dust mite extract and endotoxin instillations reduced inflammation associated vascular leakage but did not alter the transmigration of neutrophils and eosinophils into the bronchoalveolar fluid [26]. More bioavailable derivatives of compstatin, a peptide that stops C3 convertase activity, are in clinical trials of asthmatic patients [19].

In preclinical models, systemic blockade of C5 using the same mAb led to alleviation of bronchoconstriction when ovalbumin induced respiratory inflammation was established [27] and to a reduction on airway resistance without effect on leukocyte numbers in bronchoalveolar lavage fluids in response to house dust mite induced airway hypersensitivity [28]. Joint intranasal and intraperitoneal administrations of C3aR antagonist and anti-C5aR mAb led to a significant reduction of IL-4 in the bronchoalveolar lavage fluids of *Aspergillus fumigatus* stimulated mice [16]. The same C3aR antagonist, given intraperitoneally in mice reacting to ovalbumin stimulation, caused a reduction in airway resistance and of neutrophils in bronchoalveolar lavage fluids [29].

Importantly, significant engagement of the IL-17/IL-23 signaling axis in asthma produces a proinflammatory endotype which is characterized by increased mucus production, eosinophil activation, neutrophil infiltration and proliferation of smooth muscle cells [30], and increased IL-17 content in sputum from asthmatics is associated with more severe disease endotype [31]. Experimental modeling has shown that this axis is modulated by complement anaphylatoxins [32] and that the development of a Th17 response in ovalbumin induced respiratory hyperresponsiveness was significantly reduced by systemic blockade of complement properdin which reduces an inherent amplification of complement activation [33].

The application of anti-C5 mAb (eculizumab) in the clinical treatment for asthma reportedly reduced the late inflammatory response in mild allergic asthma [34]. Its mode of action may favor the decline in C5b generation over C5a production [35]. Indeed, an additional avenue independent of the activities of C3a and C5a may present itself in the form of sublytic membrane attack complexes (MAC), composed of C5b, C6, C7, C8, C9 which activate in vivo the nucleotide-binding domain and leucine-rich repeat protein 3 (NLRP3) inflammasome [36]. Upon activation, bioactive IL-1β and IL-18 are produced, with pro-inflammatory effects including neutrophil attraction and Th17 polarization. Severe, steroid-resistant and neutrophilic, asthma endotypes have been associated with increased NLRP3 expression [37], implying a possible therapeutic approach to target MAC formation or downstream signaling, i.e., inflammasome activation [38]. Preclinical gene therapy in a different disease model has shown that CD59, a GPI-anchored surface inhibitor of the complete assembly of MAC, significantly reduced NLRP3 inflammasome activation [39].

## 3. Conclusions/Way Forward

Asthma presents as varying endotypes where the development of complement therapeutics may constitute one strategic approach in the multipronged specific targeting in a subgroup of patients. The development of clinically usable tools may be desirable that influence a complement modulated tolerogenic immune phenotype in those susceptible [40]. Further preclinical (tailored to the asthmatic endotype) and clinical studies are warranted [41].

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
