# Peer review of "The Value of Targeting Complement Components in Asthma"

_medicina, 2020, doi:10.3390/medicina56080405_

Round 1
Reviewer 1 Report
This manuscript submitted by Mohamed et al. described the perspective for the value of complement therapeutics in bronchial asthma. In this manuscript, authors review the role of complement in both the immediate phase and the delayed phase of bronchial asthma and the possibility of complement for therapeutic target in bronchial asthma.
This manuscript is well described and straightforward. I think however that there are a few improvements that should be made before publication.
My major comments are as follows.
1. It is better to make a reference to the role of complement in innate immunity, even if there is no report showing positive results. This manuscript reports that complement has the important role in both Th2 response and Th 17 response in bronchial asthma. These are the acquired immunity related with the mechanism of bronchial asthma. On the other hand, the innate immunity is receiving a lot of attention in the mechanism of bronchial asthma.
Author Response
Thank you for the comment. We have addressed the point in the second paragraph of the background (section 1). Changes made in response to reviewers’ comments are highlighted in yellow.
Reviewer 2 Report
Authors discuss the potential clinical value of targeting complement in asthma.
Apart from one paper concerning the effect of C1-inhibitor in adults with mild asthma: (Allergy 2020) the review did not contain any new information compared to the review provided by Khan and Coll. (Complement components as potential therapeutic targets for asthma treatment, Respiratory Med 2014).
Authors should update their review. Jack Young and Coll (Respiratory Res 2019) aimed to determine the effect of C5 blockade during the effector phase on the pulmonary TH2 response and AHR in a house dust mite (HDM) driven murine asthma model. Anti-C5 did not affect innate lymphoid cell (ILC) proliferation or group 2 ILC (ILC2) differentiation. Anti-C5 attenuated HDM induced AHR in the absence of an effect on lung histopathology, mucus production or vascular leak.
Minor point: the title is not clear. “complement therapeutics “ should be changed in “of targeting complement components”
Author Response
Thank you for the comment. We have updated the scope of literature reviewed in sections 2 and 3. We were also more specific in the kinds of mouse stimulations conducted as we were asked to include these critically. We have changed the title as suggested.
Changes made in response to reviewers’ comments are highlighted in yellow.
Round 2
Reviewer 2 Report
The revised manuscript has been changed accordingly to criticisms and suggestions and it appears significantly improved.